# Association between *NF-kB* polymorphism and age-related macular degeneration in a high-altitude population

**Yan Xin[1]◉, Kang Zefeng[2]◉\*, Li Ling[3]◉, Guan Ruijuan [3]◉\***

**1** Medical College of Qinghai University, Xining, Qinghai Province, China, **2** Ophthalmic Hospital, Chinese Academy of Traditional Chinese Medicine, Beijing, China, **3** Department of Ophthalmology, Qinghai Provincial People's Hospital, Xining, Qinghai Province, China

◉ These authors contributed equally to this work.
\* grjchuer@163.com (GR); zefeng2531@163.com (KZ)

**Data Availability Statement:** All relevant data are within the paper and its Supporting Information files.

**Funding:** Ruijuan Guan; Xin Yan:13279497309 Zefeng Kang: 13552597717; Ling Li:

## Abstract

### Objective

To investigate the association between the nuclear factor kappa B (*NF-kB*) gene polymorphism and age-related macular degeneration (AMD) in a high-altitude population.

### Methods

Fifty-five patients with AMD and 57 control subjects were recruited from the Qinghai Provincial People's Hospital, China. Genomic DNA was extracted from the blood sample of each participant. Four *NF-kB* polymorphisms (rs3774959, rs3774932, rs3774937, and rs230526) were genotyped using a MassARRAY system. The genotype and allele frequencies were compared between the case and control groups using the chi-squared test or Fisher's exact test.

### Results

There was no significant difference in sex, age, hypertension, diabetes, blood lipid level or smoking and drinking status between the AMD and control groups ($P > 0.05$). The genotype distributions of four *NF-kB* polymorphisms were in accordance with Hardy-Weinberg equilibrium in the control group ($P > 0.05$). The frequencies of genotype AA of rs3774932 and genotype CC of rs3774937 were nominally significantly higher in the AMD group than in the control group ($P = 0.046$ and $0.023$, respectively), although these associations did not survive the Bonferroni correction (corrected $P > 0.05$). Genotype distributions of rs3774959 and rs230526 were not significantly different between the two groups ($P = 0.08$ and $0.16$, respectively). No significant difference in the allele frequencies of the four polymorphisms was found between the AMD and control groups ($P > 0.05$).

### Conclusions

Genotype AA of rs3774932 and genotype CC of rs3774937 in *NF-kB* might be risk factors for AMD.

13309712948, Ruijuan Guan: 18997150572;
Department of Science and Technology of Qinghai
Province.

**Competing interests:** The authors have declared
that no competing interests exist.

## Introduction

Age-related macular degeneration (AMD) is one of the most common irreversible blinding
eye diseases, which seriously affects patients' quality of life. The disease affects tens of millions
of people all over the world [1]. In China, the prevalence of AMD ranged from 2.44% in people
aged 45–49 years to 18.98% in people aged 85–89 years [2]. The development of AMD is
related to many factors, such as age, sex, race, eating habits, obesity, and sun exposure, but the
specific etiology is not clear [3]. Among various factors affecting the pathogenesis of AMD,
mitochondrial damage in the retinal pigment epithelium (RPE) is an important cause of RPE
dysfunction [4]. Studies have shown that mitochondrial damage in the RPE is closely related
to inflammation, autophagy, and apoptosis [5–7]. In recent years, results from many studies
have suggested that genetic factors may play an important role in the development of AMD.
Studies have shown that the heritability of AMD can be as high as 71% [8], which indicates
that genetic factors are closely related to the development of AMD. Over 50% of the heritability
of AMD has been explained by two major genes (CFH and ARMS2/HTRA1), making it one of
the most well-defined genetically complex disorders [9]. A recent genome-wide association
study (GWAS) on 43,566 subjects identified 52 independently associated variants spanning 34
loci for AMD [10].

The nuclear factor kappa B (*NF-kB*) family of transcription factors plays a pivotal role in
regulating inflammatory response, immune function, and malignant transformation [11]. In
addition, *NF-kB* affects the expression of genes for cell [6] differentiation, proliferation, and
survival in almost all multicellular organisms [12]. There is also a significant correlation
between *NF-kB* and autophagy [13]. *NF-kB* is composed of a group of homodimer and hetero-
dimer protein complexes, and p50/p65/p53 heterodimer complex is the most common com-
plex [14]. Results from studies have suggested that the *NF-kB* gene plays a role in the
occurrence of lung cancer, colorectal cancer, breast cancer, and other cancers [15–18]. Some
studies have shown that the *NF-kB* gene has strong correlation with inflammation and autop-
hagy, while inflammation and autophagy are important pathogenesis of AMD [19–23]. A
recent genome-wide meta-analysis identified novel loci associated with AMD, including
*C4BPA-CD55*, *ZNF385B*, *ZBTB38*, and *NFKB1* [24].

However, researchers have not investigated the association between the *NF-kB* gene and the
AMD risk in a Chinese population. In the present study, four single nucleotide polymorphisms
(SNPs) of *NF-kB* (rs3774959, rs3774932, rs3774937, and rs230526) were selected for genotyp-
ing and their associations with the risk of AMD were analyzed in a high-altitude Chinese
population.

## Methods

### Study participants

Fifty-five patients with AMD were and 57 control participants were recruited from the Qing-
hai Provincial People's Hospital, China from December 2016 to December 2019. This study
was conducted in accordance with the tenets of the Declaration of Helsinki and has been
approved by the local hospital ethics committee (approval number 2017–21). Written
informed consent was obtained from all participants.

According to the *Preferred Practice Pattern Guidelines: Age-related Macular Degeneration*
[25] and the "Chinese clinical diagnosis and treatment pathway for age-related macular degen-
eration" [26], AMD was diagnosed when the presence of one or more of the following criteria
was met: 1) medium-sized hyaline warts (>63 μm in diameter); 2) RPE abnormalities such as
hypopigmentation, pigment proliferation, migration, and metaplasia; and 3) any of the

following characteristics: retinal pigment epithelium map atrophy, choroidal neovascularization (exudative), polypoid choroidal vasculopathy, and retinal hemangioma hyperplasia.

The inclusion criteria for AMD patients were as follows: 1) long-term (>20 years) residence at altitudes >2,000 m; 2) age >40 years; and 3) diagnosed with AMD based on the diagnostic criteria. The exclusion criteria for AMD patients were as follows: 1) late vitelliform macular degeneration; 2) choroidal neovascularization with high myopia; 3) Stargardt disease; 4) retinal vascular occlusion; 5) chorioretinitis; 6) diabetic retinopathy; 7) hypertensive retinopathy; and 8) other eye diseases.

All control participants were >40 years of age and diagnosis of AMD and other fundus diseases had been ruled out by fundus examination. Participants with other diseases such as high myopia and glaucoma were also excluded. In addition, those with serious systemic diseases such as hypertension, diabetes, renal insufficiency, blood diseases, and benign or malignant tumors were excluded.

## Sample collection and DNA extraction

Two milliliters of peripheral blood was collected from all participants. After anticoagulant treatment, the samples were frozen at −8°C before use. Genomic DNA was extracted using xxx. DNA quality was assessed using the NanoDrop 2000 spectrophotometer (Thermo Fisher Scientific, Waltham, MA, USA).

## SNP genotyping

Single nucleotide polymorphisms (SNPs) were genotyped using a MassARRAY system (Sequenom, San Diego, CA, USA). Single-base extension primers were designed and synthesized by the Sangon Biotech (Shanghai, China). The sequences of the primers are listed in Table 1.

The PCR reaction was performed using a MassARRAY mass spectrometer (Sequenom, San Diego, CA). Genotypes were called using the MassARRAY RS1000 software (Sequenom).

## Statistical analysis

SPSS statistical software (version 25.0; IBM, Armonk, NY, USA) was used for statistical analysis. The numerical data were expressed as mean ± standard deviation and compared between the case and control groups using the Student's *t* test. The categorical data were expressed as n (%) and compared between the two groups using the chi-squared test. Hardy-Weinberg equilibrium was tested for genotype distributions using the chi-squared test. Genotype and allele frequencies were compared between the case and control groups using the chi-squared test or Fisher's exact test. We corrected for multiple tests using the Bonferroni method. A $P < 0.05$ was considered statistically significant. Power analysis was performed using the Genetic Power Calculator [27].

**Table 1. Sequences of the PCR primers used in this study.**

| SNP | Forward primer | Reverse primer | Amplicon (bp) |
|---|---|---|---|
| rs3774959 | 5'-AGTAACACCACATAGGCAGTAACG-3' | 5'-TGACTGATGAGATACTGGGGCTA-3' | 184 |
| rs3774932 | 5'-TCTAGCAGAATCCCACAACTGAATA-3' | 5'-ATTCCAAGTCTCCATTAATCGTACA-3' | 332 |
| rs3774937 | 5'-CATAATGCATGAGGTCTACTTCTTC-3' | 5'-AACTGCTTCAATACCTCTGTTCATG-3' | 242 |
| rs230526 | 5'-GCTTGGCAGCAGCAATTTAA-3' | 5'-TACGGCAACAAAGGTACATACAA-3' | 307 |

PCR, polymerase chain reaction; SNP, single nucleotide polymorphism.

**Table 2. Demographic and clinical features of patients with AMD and controls.**

| Feature | AMD (n = 55) | Controls (n = 57) | *P* |
|---|---|---|---|
| Female (%) | 27 (49.1) | 28 (49.1) | 0.99 |
| Age (years) | 66.5 ± 11.8 | 65.5 ± 11.5 | 0.13 |
| Hypertension (%) | 21 (38.2) | 19 (33.3) | 0.69 |
| Diabetes (%) | 7 (12.7) | 9 (15.8) | 0.98 |
| Hyperlipidemia (%) | 9 (16.4) | 7 (12.3) | 0.54 |
| Smoking (%) | 4 (7.3) | 6 (10.5) | 0.55 |
| Drinking (%) | 4 (7.3) | 6 (10.5) | 0.55 |

AMD, age-related macular degeneration.

## Results

### Demographic and clinical features of patients with AMD and controls

The AMD group consisted of 28 males and 27 females, with an average age of 66.5 (±11.8) years. The control group had 29 males and 28 females, with an average age of 65.5 (±11.5) years. There was no significant difference in sex, age, hypertension, diabetes, blood lipid level, or smoking or drinking status between the AMD group and the control group ($P > 0.05$; Table 2).

### Genotype distributions of *NF-kB* gene polymorphisms in patients with AMD and controls

Genotype distributions of all four SNPs followed Hardy-Weinberg equilibrium in the control group ($P > 0.05$). The frequencies of genotype AA of rs3774932 and genotype CC of rs3774937 were nominally significantly higher in the AMD group than in the control group ($P = 0.046$ and $0.023$, respectively), although these associations did not survive the Bonferroni correction (corrected $P > 0.05$). Genotype distributions of rs3774959 and rs230526 were not significantly different between the two groups ($P = 0.08$ and $0.16$, respectively; Table 3).

### Allele distributions of *NF-kB* gene polymorphisms in patients with AMD and controls

There was no significant difference in allele frequencies of four polymorphisms between the AMD and control groups ($P > 0.05$; Table 4).

**Table 3. Genotype distributions of *NF-kB* gene polymorphisms in patients with AMD and controls.**

| SNP | Allele | Group | Genotype, n (%) | | | *P* |
|---|---|---|---|---|---|---|
| | (1/2) | | 1/1 | 1/2 | 2/2 | |
| rs3774959 | G/A | AMD | 22 (0.400) | 21 (0.382) | 12 (0.218) | 0.08 |
| | | Control | 26 (0.456) | 27 (0.474) | 4 (0.070) | |
| rs3774932 | A/G | AMD | 21 (0.382) | 19 (0.345) | 15 (0.273) | 0.046 |
| | | Control | 14 (0.246) | 33 (0.579) | 10 (0.175) | |
| rs3774937 | T/C | AMD | 29 (0.527) | 15 (0.273) | 11 (0.200) | 0.023 |
| | | Control | 28 (0.491) | 26 (0.456) | 3 (0.053) | |
| rs230526 | G/A | AMD | 19 (0.345) | 23 (0.418) | 13 (0.236) | 0.16 |
| | | Control | 20 (0.351) | 31 (0.544) | 6 (0.105) | |

AMD, age-related macular degeneration; SNP, single nucleotide polymorphism.

**Table 4. Genotype distributions of *NF-kB* gene polymorphisms in patients with AMD and controls.**

| SNP | Allele | AMD group | Control group | P | OR | 95% CI |
|---|---|---|---|---|---|---|
| rs3774959 | A | 45 (0.409) | 35 (0.307) | 0.11 | 1.56 | [0.90, 2.71] |
| | G | 65 (0.591) | 79 (0.693) | | | |
| rs3774932 | A | 61 (0.555) | 61 (0.535) | 0.77 | 1.08 | [0.64, 1.83] |
| | G | 49 (0.445) | 53 (0.465) | | | |
| rs3774937 | C | 37 (0.336) | 32 (0.281) | 0.37 | 1.30 | [0.74, 2.29] |
| | T | 73 (0.664) | 82 (0.719) | | | |
| rs230526 | A | 49 (0.445) | 43 (0.377) | 0.30 | 1.33 | [0.78, 2.26] |
| | G | 61 (0.555) | 71 (0.623) | | | |

AMD, age-related macular degeneration; CI, confidence interval; OR, odds ratio; SNP, single nucleotide polymorphism.

## Discussion

AMD is a complex, highly inherited multifactorial disease caused by the interaction of genetic and environmental risk factors [28]. In this study, we compared the frequencies of *NF-kB* gene polymorphisms between AMD cases and controls to explore the correlation between the *NF-kB* gene and AMD. The allele frequencies of the four SNPs analyzed in this study are comparable to the corresponding allele frequencies in East Asians reported in the public gnomAD database [29] (Table 4). Our results showed that the frequencies of genotype AA of rs3774932 and genotype CC of rs3774937 were nominally significantly higher in the AMD group than in the control group (Table 3), although these associations did not survive the Bonferroni correction (corrected $P > 0.05$). These findings suggest that individuals carrying genotype AA of rs3774932 or genotype CC of rs3774937 may have a higher risk of developing AMD.

Silico analyses suggest that SNP rs3774932 could change the Nkx2 and SIX5 motifs of the *NF-kB* protein, while SNP rs3774937 could change the DMRT1, LUN-1, and YY1 motifs of the *NF-kB* protein [30]. Functional studies such as the ChIP assay to identify the binding activity of these genotypes or luciferase reporter assay to test the function of these polymorphisms, especially the *NF-kB* binding site activity in genotype AA of rs3774932 or genotype CC of rs3774937, would be helpful to further elucidate the potential effects of these genotypes on gene transcriptional regulation and expression, consequently affecting the *NF-kB* pathway activation and/or susceptibility to AMD pathology.

This study had several limitations. First, the power of this study to detect the association between the *NF-kB* gene polymorphisms and AMD was limited due to small sample sizes. We estimated that this study had a power of 0.78 to detect the association at OR = 2.0 and a prevalence of 7.11% for people aged 65–69 years [2]. Therefore, the negative associations we observed after correcting for multiple tests in this study did not disprove potential real associations between the *NF-kB* gene polymorphisms and AMD. Second, the use of 40 years of age as an inclusion criterion for AMD cases and controls may have compromised the representativeness of the samples in this study. Although AMD can be diagnosed as early as 35 years, most AMD cases are diagnosed at age 60 years and older [2]. In addition, since control participants younger than 60 years may develop AMD in their later life, the age cut-off may have introduced selection bias in this study, resulting in lower power to detect real associations. Third, even though one of the strengths of this study is that all patients with AMD were long-term residents (>20 years) at a high altitude (over 2,000 m), which is a unique population that has been under-represented in genetic studies of AMD, the lack of comparison to non-high-altitude patients in this study made it impossible to conclude that the risk alleles are associated with AMD in the high-altitude population alone.

In conclusion, the results of this study suggest that the AA genotype at rs3774932 and the CC genotype at rs3774937 in the *NF-kB* gene may be risk factors for the development of AMD. This is the first study that indicates an association between these two NF-kB variants and the risk of AMD. Whether this association exists in non-Chinese populations is worth further study.

## Supporting information

**S1 File.**
(PDF)

## Author Contributions

**Data curation:** Yan Xin.

**Formal analysis:** Yan Xin.

**Funding acquisition:** Li Ling, Guan Ruijuan.

**Methodology:** Guan Ruijuan.

**Software:** Yan Xin.

**Supervision:** Kang Zefeng, Li Ling, Guan Ruijuan.

**Writing – original draft:** Yan Xin.

**Writing – review & editing:** Yan Xin.

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
