## [Decision Letter · Decision Letter 0]

4 Mar 2021

PONE-D-20-40186

Association between NF-kB polymorphism and Age-related macular degeneration in high altitude populationAssociation between NF-kB polymorphism and A ge-related macular degeneration in high altitude population

PLOS ONE

Dear Dr. Guan,

Thank you for submitting your manuscript to PLOS ONE. After careful consideration, we feel that it has merit but does not fully meet PLOS ONE’s publication criteria as it currently stands. Therefore, we invite you to submit a revised version of the manuscript that addresses the points raised during the review process.

Authors are required to address all the issues raised by both reviewers. Specifically, the authors need to clarify the use of age 40 for AMD and ethical committee number. Additionally, language needs to be edited and spelling and grammar to improve the clarity for the reader.

We look forward to receiving your revised manuscript.

Kind regards,

Ravirajsinh Jadeja, Ph.D

Academic Editor

PLOS ONE

Journal Requirements:

2. PLOS ONE has specific criteria requiring that experiments, statistics, and other analyses are performed to a high technical standard and are described in sufficient detail (https://journals.plos.org/plosone/s/criteria-for-publication#loc-3). To that effect, please ensure that you have described who performed the methods and procedures and a more thorough discussion of how they were carried out, as well as further details regarding participant recruitment. If materials, methods, and protocols are well established, authors may cite articles where those protocols are described in detail, but your submission should include sufficient information to be understood independent of these references (https://journals.plos.org/plosone/s/submission-guidelines#loc-materials-and-methods).

*Please note that PLOS ONE does not copy edit accepted manuscripts (https://journals.plos.org/plosone/s/criteria-for-publication#loc-5). To that effect, please ensure that your submission is free of typos and grammatical errors.

"I have read the journal's policy and the authors of this manuscript have the following competing interests: [insert competing interests here]"

4. Please include your tables as part of your main manuscript and remove the individual files. Please note that supplementary tables (should remain/ be uploaded) as separate "supporting information" files

5. We note you have included a table to which you do not refer in the text of your manuscript. Please ensure that you refer to Table 2 in your text; if accepted, production will need this reference to link the reader to the Table.

Reviewers' comments:

Reviewer's Responses to Questions

**Comments to the Author**

1. Is the manuscript technically sound, and do the data support the conclusions?

Reviewer #1: No

Reviewer #2: No

2. Has the statistical analysis been performed appropriately and rigorously? 

Reviewer #1: No

Reviewer #2: Yes

3. Have the authors made all data underlying the findings in their manuscript fully available?

Reviewer #1: Yes

Reviewer #2: Yes

4. Is the manuscript presented in an intelligible fashion and written in standard English?

Reviewer #1: No

Reviewer #2: Yes

5. Review Comments to the Author

Reviewer #1: Yin et al examined 55 patients and 57 controls from the Qinghai Provicial People’s Hospital and genotyped them for four SNPs in NFK-B. The paper claims a risk association with NFK-B for risk of AMD in a population that had lived more than 20 years in a high altitude climate (>2000m). The study is interesting but there are some major issues.

1) The usual age of diagnosis of AMD is >60. However, here >age 40 was used. Could the authors please clarify?

2) The sample size is extremely small (slightly over 100 people), and therefore conclusions regarding age, gender, or other differentiation is extremely hard to ascertain, including association of NFK-B to AMD. Could the authors provide please how they were able to have the power to draw the conclusions they did?

3) No comparison to non-high altitude patients was obtained, even a larger study to evaluate the NFK-B polymorphism to see how common the alleles are in the regular population- which means that this study cannot come to the conclusion as to whether or not the alternate allele is associated with AMD in the high-altitude population alone. The reader does not even know how common the alternate alleles are in the general population through examining databases like GnomeAD or ExAC. Could the authors please clarify?

4) The p-values are extremely high (0.04) to declare an association.

5) The paper language needs to be edited along with spelling and grammar, to improve the clarity for the reader.

Reviewer #2: Association between NF-kB polymorphism and Age-related macular degeneration in high altitude population

The authors have intended to investigate the association between NF-kB gene polymorphism and age-related macular degeneration in high altitude population. The study concludes that AA genotype at rs3774932 and CC genotype at rs3774937 in NF-kB gene could be the risk genotypes for AMD. However, this reviewer has few major concerns to be addressed by the authors in order to enhance the quality of the manuscript.

• The ethical committee numbers are different on the ethic statement and in the manuscript.

Introduction: Overall, this section needs to be revised.

• The details in this section is not sufficient enough to substantiate the primary goal. I suggest including more information and citations relevant to Nf-KB gene and its association with AMD.

• The subheading “Subjects” could be under the methods section.

Methods: Overall, the method section needs a revision and more references could be added to enable the readers to follow without confusion.

• 1.2.1 - Provide sample size with more details.

• 1.2.2 - Provide reference and add more details.

• 1.2.3 - Provide the primers and probe details in the table format.

• 1.2.4 - The representative gel images could be added.

• 1.2.5 - very less details

• Did the author further confirm the genotypes/SNP of these four variants of NF-KB by PCR-RFLP?

• Besides, either the ChIP assay to identify the binding activity of this genotypes or luciferase reporter assay to test the function of these polymorphisms, especially the NF-κB binding site activity in AA and CC alleles, could have been attempted.

Results: Overall, the results are not presented well and need thorough revision.

• 2.3 - grou-----group

Discussion:

The current data obtained from this study was not sufficient enough to conclude. Also, the findings are not discussed well.

• The results should be discussed with respect to how these genetic variations (rs3774932 and rs3774937) might influence gene transcriptional regulation and expression, consequently affecting NF-κB pathway activation and/or the susceptibility to AMD pathology.

Figures &Tables:

Some of the details on table-3 and supplementary materials are provided in different language.

Please consider.

• The punctuations/ spacings throughout the manuscript should be rectified.

6. PLOS authors have the option to publish the peer review history of their article (what does this mean?). If published, this will include your full peer review and any attached files.

Reviewer #1: No

Reviewer #2: No

---

## [Author Response · Author response to Decision Letter 0]

9 Apr 2021

Response: We have revised our manuscript to meet PLOS ONE's style requirements.

2. PLOS ONE has specific criteria requiring that experiments, statistics, and other analyses are performed to a high technical standard and are described in sufficient detail (https://journals.plos.org/plosone/s/criteria-for-publication#loc-3). To that effect, please ensure that you have described who performed the methods and procedures and a more thorough discussion of how they were carried out, as well as further details regarding participant recruitment. If materials, methods, and protocols are well established, authors may cite articles where those protocols are described in detail, but your submission should include sufficient information to be understood independent of these references (https://journals.plos.org/plosone/s/submission-guidelines#loc-materials-and-methods).

*Please note that PLOS ONE does not copy edit accepted manuscripts (https://journals.plos.org/plosone/s/criteria-for-publication#loc-5). To that effect, please ensure that your submission is free of typos and grammatical errors.

Response: We have revised the Methods according to PLOS ONE's requirements.

"I have read the journal's policy and the authors of this manuscript have the following competing interests: [insert competing interests here]"

Response: We have added the statement "The authors have declared that no competing interests exist" to the manuscript.

4. Please include your tables as part of your main manuscript and remove the individual files. Please note that supplementary tables (should remain/ be uploaded) as separate "supporting information" files

Response: We have included all tables as part of the main manuscript and have removed the individual files.

5. We note you have included a table to which you do not refer in the text of your manuscript. Please ensure that you refer to Table

Response: All tables have been referred to within the text of the revised manuscript.

Responses to Reviewer #1:

Reviewer #1: Yin et al examined 55 patients and 57 controls from the Qinghai Provincial People’s Hospital and genotyped them for four SNPs in NFK-B. The paper claims a risk association with NFK-B for risk of AMD in a population that had lived more than 20 years in a high-altitude climate (> 2000 m). The study is interesting but there are some major issues.

Response: Thank you for the opportunity to improve our manuscript.

1) The usual age of diagnosis of AMD is > 60. However, here > age 40 was used. Could the authors please clarify?

Response: We agree with the Reviewer that the use of ≥ 40 years of age as an inclusion criterion for AMD cases and controls may have compromised the representativeness of the samples in this study. Although AMD can be diagnosed as early as 35 years, most AMD cases are diagnosed at ≥ 60 years [Song P, Du Y, Chan KY, Theodoratou E, Rudan I. The national and subnational prevalence and burden of age-related macular degeneration in China. J Glob Health. 2017 Dec;7(2):020703.]. In addition, since control participants younger than 60 years may develop AMD later in life, our inclusion criterion of patients ≥ 40 years may have introduced selection bias in this study, resulting in lower power to detect real associations. We have added these statements to the Discussion.

2) The sample size is extremely small (slightly over 100 people), and therefore conclusions regarding age, gender, or other differentiation are extremely hard to ascertain, including association of NFK-B to AMD. Could the authors provide please how they were able to have the power to draw the conclusions they did?

Response: We agree with the Reviewer that the power of our study to detect the association between the NF-kB gene polymorphisms and AMD was limited due to the small sample size. We estimated that this study had the power of 78% to detect the association at OR = 2.0 and the prevalence at 7.11% for people aged 65–69 years [Song P, Du Y, Chan KY, Theodoratou E, Rudan I. The national and subnational prevalence and burden of age-related macular degeneration in China. J Glob Health. 2017 Dec;7(2):020703.]. Therefore, the negative associations we observed after correcting for multiple tests in this study did not disprove potential associations between the NF-kB gene polymorphisms and AMD. We have added these statements to the Discussion.

3) No comparison to non-high-altitude patients was obtained, even a larger study to evaluate the NFK-B polymorphism to see how common the alleles are in the regular population- which means that this study cannot come to the conclusion as to whether or not the alternate allele is associated with AMD in the high-altitude population alone. The reader does not even know how common the alternate alleles are in the general population through examining databases like GnomeAD or ExAC. Could the authors please clarify?

Response: One of the strengths of this study is that all patients with AMD were long-term (> 20 years) residents at a high altitude (> 2,000 m), which is a unique population that is under-represented in genetic studies of AMD. However, we agree that the lack of comparison to non-high-altitude patients in this study made it impossible to conclude that the risk alleles are associated with AMD in the high-altitude population alone. The allele frequencies of the four SNPs analyzed in this study are comparable to the corresponding allele frequencies in East Asians reported in the public gnomAD database (Table 3). We have added these statements to the Discussion.

4) The p-values are extremely high (0.04) to declare an association.

Response: We agree that the associations we observed in this study were nominally significant and did not survive the Bonferroni correction (corrected P > 0.05). We have clarified this in the Results.

5) The paper language needs to be edited along with spelling and grammar, to improve the clarity for the reader.

Response: Thank you for the comments. We have improved our language by correcting spelling and grammar.

Responses to Reviewer #2:

Reviewer #2: Association between NF-kB polymorphism and age-related macular degeneration in a high-altitude population.

The authors have intended to investigate the association between NF-kB gene polymorphism and age-related macular degeneration in a high-altitude population. The study concludes that the AA genotype at rs3774932 and CC genotype at rs3774937 in NF-kB gene could be the risk genotypes for AMD. However, this reviewer has few major concerns to be addressed by the authors in order to enhance the quality of the manuscript.

• The ethical committee numbers are different on the ethics statement and in the manuscript.

Response: Thank you for pointing out this error. We have corrected it in the manuscript.

Introduction: Overall, this section needs to be revised.

Response: We have revised the Introduction.

• The details in this section are not sufficient enough to substantiate the primary goal. I suggest including more information and citations relevant to the Nf-KB gene and its association with AMD.

Response: We have added more information and citations relevant to the NF-kB gene and its association with AMD.

• The subheading “Subjects” could be under the methods section.

Response: We have revised the subheadings across the entire manuscript.

Methods: Overall, the methods section needs a revision and more references could be added to enable the readers to follow without confusion.

• 1.2.1 - Provide sample size with more details.

• 1.2.2 - Provide reference and add more details.

• 1.2.3 - Provide the primers and probe details in the table format.

• 1.2.4 - The representative gel images could be added.

• 1.2.5 - Very less details

Response: We have revised the Methods according to the Reviewer’s comments.

• Did the author further confirm the genotypes/SNP of these four variants of NF-KB by PCR-RFLP?

Response: We did not further confirm the genotypes of these four SNPs in NF-kB by PCR-RFLP. However, we used the well-known MassARRAY system to genotype the SNPs.

• Besides, either the ChIP assay to identify the binding activity of these genotypes or luciferase reporter assay to test the function of these polymorphisms, especially the NF-κB binding site activity in AA and CC alleles, could have been attempted.

Response: Thank you for the comments on functional studies. Those will be one of the major aims in our future studies.

Results: Overall, the results are not presented well and need thorough revision.

Response: We have thoroughly revised the Results section.

• 2.3 - grou-----group

Response: We have corrected this typo.

Discussion:

The current data obtained from this study was not sufficient enough to conclude. Also, the findings are not discussed well.

• The results should be discussed with respect to how these genetic variations (rs3774932 and rs3774937) might influence gene transcriptional regulation and expression, consequently affecting NF-κB pathway activation and/or the susceptibility to AMD pathology.

Response: We have revised the Discussion.

Figures &Tables:

Some of the details on table-3 and supplementary materials are provided in a different language.

Response: We have removed all non-English text from the manuscript.

Please consider.

• The punctuations/ spacings throughout the manuscript should be rectified.

Response: We have reformatted the entire manuscript.

---

## [Editor Report · Decision Letter 1]

6 May 2021

Association between NF-kB polymorphism and age-related macular degeneration in a high-altitude population

PONE-D-20-40186R1

Dear Dr. Guan,

We’re pleased to inform you that your manuscript has been judged scientifically suitable for publication and will be formally accepted for publication once it meets all outstanding technical requirements.

Kind regards,

Ravirajsinh Jadeja, Ph.D

Academic Editor

PLOS ONE
---

## [Editor Report · Acceptance letter]

14 May 2021

PONE-D-20-40186R1 

Association between *NF-kB* polymorphism and age-related macular degeneration in a high-altitude population 

Dear Dr. Ruijuan:

I'm pleased to inform you that your manuscript has been deemed suitable for publication in PLOS ONE. Congratulations! Your manuscript is now with our production department. 

Kind regards, 

on behalf of

Dr. Ravirajsinh Jadeja 

Academic Editor

PLOS ONE